# Uncovering the Gene Regulatory Network of Maize Hybrid ZD309 under Heat Stress by Transcriptomic and Metabolomic Analysis

**DOI:** 10.3390/plants11050677

**Published:** 2022-03-01

**Authors:** Jingbao Liu, Linna Zhang, Lu Huang, Tianxiao Yang, Juan Ma, Ting Yu, Weihong Zhu, Zhanhui Zhang, Jihua Tang

**Affiliations:** 1Institute of Cereal Crops, Henan Academy of Agricultural Sciences, Zhengzhou 450002, China; jbliu1777@126.com (J.L.); zhanglinna21@163.com (L.Z.); huanglu96@126.com (L.H.); majuanjuan85@126.com (J.M.); yuting971117@163.com (T.Y.); lzszhuweihong@163.com (W.Z.); 2Plant Molecular and Cellular Biology Program, University of Florida, Gainesville, FL 32611, USA; tianxiao.yang@ufl.edu; 3National Key Laboratory of Wheat and Maize Crop Science, Collaborative Innovation Center of Henan Grain Crops, College of Agronomy, Henan Agricultural University, Zhengzhou 450002, China

**Keywords:** maize (*Zea mays* L.), hybrid, heat tolerance, transcriptome, metabolome, gene regulatory network, grain yield, physical alterations

## Abstract

Maize is an important cereal crop but is sensitive to heat stress, which significantly restricts its grain yield. To explore the molecular mechanism of maize heat tolerance, a heat-tolerant hybrid ZD309 and its parental lines (H39_1 and M189) were subjected to heat stress, followed by transcriptomic and metabolomic analyses. After six-day-heat treatment, the growth of ZD309 and its parental lines were suppressed, showing dwarf stature and rolled leaf compared with the control plants. ZD309 exhibited vigorous growth; however, M189 displayed superior heat tolerance. By transcriptomic and metabolomic analysis, hundreds to thousands of differentially expressed genes (DEGs) and metabolites (DEMs) were identified. Notably, the female parent H39 shares more DEGs and DEMs with the hybrid ZD309, indicating more genetic gain derived from the female instead of the male. A total of 299 heat shock genes detected among three genotypes were greatly aggregated in sugar transmembrane transporter activity, plasma membrane, photosynthesis, protein processing in the endoplasmic reticulum, cysteine, and methionine metabolism. A total of 150 heat-responsive metabolites detected among three genotypes were highly accumulated, including jasmonic acid, amino acids, sugar, flavonoids, coumarin, and organic acids. Integrating transcriptomic and metabolomic assays revealed that plant hormone signal transduction, cysteine, and methionine metabolism, and α-linolenic acid metabolism play crucial roles in heat tolerance in maize. Our research will be facilitated to identify essential heat tolerance genes in maize, thereby contributing to breeding heat resistance maize varieties.

## 1. Introduction

High temperature as a consequence of global warming has become a common concern worldwide that threatens to crop production [1,2,3]. Maize (*Zea mays* L.) is a well-known important crop for food, fuel, and feed [4] but is sensitive to heat stress, decreasing grain yield [2,5]. Prior data have shown that with an average temperature increase of 1 °C, maize grain yield decreased by an average of 7.4% [6]. Heat stress causes additional negative effects on maize growth and development, such as dwarf plant stature, reduced pollen viability, and poor seed set [2,3,5,7]. Adaptation to temperature gradients is the primary restriction of maize to growth around the world, regardless of latitude and altitude [4,8,9].

In plants, the network of heat stress response involves heat shock transcription factor (HSF), heat shock protein (HSP), reactive oxygen species (ROS), and phytohormones pathways [2,3,10,11]. Heat stress triggers heat-response cascade, as it activates the expression of genes encoding HSPs and other related proteins and promotes the formation of HSPs, antioxidants, hormones, and other osmoprotectants to counteract heat stress-induced physio-biochemical changes [3,12]. Upon the sophisticated pathway, HSPs, HSFs, and protein kinases play essential roles in the heat-response signaling, thus protecting cellular compartments from over-heating temperature [10]. The signal cascade initiates terminal, calmodulin, or protein kinase regulates HSFs, followed by HSFs-guided transcriptional regulation, and consequently modulates downstream heat stress response genes [13]. Jasmonic acid is well-documented in biotic stress response regulation [2], for example, the defense of viral. However, current results also found that jasmonic acid is involved in the protective response against heat stress [14,15]. The molecular mechanism underlying heat stress response has been largely uncovered, and the gene regulatory network is still less addressed.

Recently, increasing research has suggested the heat stress effect on metabolome profile in wheat, rice, and maize [16,17,18,19,20]. Several heat stress-related metabolites were identified, such as sugars, amino and organic acids, and carbohydrates-related metabolites. Three major secondary metabolites were found in response to heat stress, including alkaloids, flavonoids, and polyamines in tomatoes [21]. A combined transcriptome and metabolome strategy could better explain the transcriptional regulation of metabolic pathways. Numerous heat-responsive genes have been discovered by using this combined omics strategy [2,9,18,22]. Only a few pieces of evidence are available in maize.

In maize, heat tolerance is mainly determined by a genetic basis and can be affected by plant development and other abiotic stresses. Therefore, optimizing heat tolerance traits and developing abiotic stress-resistant varieties is waiting to solve the issue for food security. In our previous study, we bred an elite single-hybrid, ZD309 (Henan Academy of Agricultural Sciences, China, dent grain), with superior grain yield, heat stress tolerance, and suitable for mechanized harvesting. Here, transcriptomic and metabolomic analyses were performed on the hybrid ZD309 and its parental lines to uncover the regulatory network of maize heat tolerance.

## 2. Results

### 2.1. Estimation of the Effects of Heat Stress on Grain Yield

In the Huang-Huai-Hai River Basin area of China, the growth of maize is suffered from severe climate conditions, particularly extreme high temperatures. To estimate the heat stress effect on ZD309 grain yield, we selected 12 commercial hybrids to be subjected to heat stress (HS) treatment in the greenhouse (Table 1). The heat-tolerant coefficient was calculated by HS grain yield/CK grain yield. Under heat stress, kernel development was repressed in other hybrids, with almost no apparent impact on the ZD309. Within the 12 tested hybrids, ZD309, DK653, Zhongkeyu505, and Zhengdan319 exhibited superior heat tolerance. However, heat stress caused notable grain yield loss in Zhengdan1002, Zhengdan326, Pionner335, and Zhengdan958.

### 2.2. Evaluation of the Morphological and Physiological Effects of Heat Stress on Seeding Development

To identify the effects of heat stress on ZD309 seedlings, five-week-old potted planted seedlings were subjected to heat stress treatment. After six-day-heat stress, the growth of ZD309 and its parental lines were greatly suppressed (Figure 1A–C), exhibiting dwarf stature and rolled leaf in comparison to the corresponding controls. The ZD309 exhibited the most negative effects on stature, along with the least effects on M189 among three tested materials. ZD309 showed enhanced growth vigor compared with its parental lines. Inconsistent with the morphological changes, heat stress also mediated physiological changes (Figure 1D–M). Among the 10 tested physiological parameters, the proline content in ZD309 and its parental lines were increased in heat stress treatment for three days (D3HS) but decreased in heat stress treatment for six days (D6HS). The malondialdehyde (MDA, an indicator of lipid peroxidation), H_2_O_2_, and soluble sugar contents in the three tested materials were increased in D3HS treatment and sustainedly increased in D6HS treatment. The chlorophyll a contents in the three tested materials were decreased in D3HS treatment and continued to be decreased in D6HS treatment. The soluble protein contents, SOD, POD, and CAT enzyme activities of ZD309 and its parental lines were increased in both heat treatments and climbed a higher level in the D3HS treatment. The chlorophyll b levels of ZD309 and its parental lines were increased in both heat stress treatments; however, no significant differences between D3HS and D6HS treatments. In both treatments, all parameters were the highest in ZD309 but the lowest in M189. The contents of MDA and H_2_O_2_ in M189 in both treatments were the highest but the lowest in ZD309. The chlorophyll b contents showed no significant difference among the three tested materials. Overall, ZD309 and its parental lines displayed superior heat tolerance but varied from the morphological and physiological parameters.

### 2.3. DEGs of ZD309 and Its Parental Lines in Response to Heat Stress

Using the Illumina paired-end RNA-seq approach, 1228 million reads were detected, and 184.21 gigabases (Gb) data were generated. After removing the low-quality data, a total of 181.03 Gb of clean data were produced. The sequencing data have been submitted to the NCBI Short Read Archive with accession number (PRJNA791560).

The transcriptome data showed time-dependent gene expression patterns under heat treatments, and the heat stress response of ZD309 was different from those of its parental lines. The number of up-regulated DEGs is more than down-regulated DEGs at each time point in all three materials. This indicated that more genes were switched on than switched off under heat stress. In D3HS treatment, 5440 (3797 up- and 1643 down-regulated), 3200 (1667 up- and 1533 down-regulated), and 4348 (2861 up- and 1487 down-regulated) DEGs were identified within the heat-stressed H39_1, M189, ZD309, and their controls, respectively (Figure 2A). In D6HS treatment, 3921 (2815 up- and 1106 down-regulated), 2424 (1486 up- and 938 down-regulated), and 2391 (1266 up- and 1125 down-regulated) DEGs were identified within the heat-stressed H39_1, M189, ZD309, and their controls, respectively (Figure 2B). In D3HS treatment, 1259 common DEGs were found among the three tested materials (Figure 2A). Moreover, 2805 genes were overlapped between the inbred line H39_1 and the hybrid ZD309, 1772 genes were overlapped between M189 and ZD309, as well as 1667 genes were overlapped between H39_1 and M189. In D6HS treatment, 614 common DEGs were found (Figure 2B). A total of 1129, 1029, and 1028 genes were shared between H39_1 and ZD309, M189, and Z309, as well as H39_1 and M189, respectively. Compared with the D6HS treatment, more DEGs were detected in ZD309 and its parental lines in the D3HS treatment. This suggested that ZD309 and its parental lines start to adapt to the stress as the treatment goes on. Overlapped DEGs between ZD309 and H39_1 were more than those of M189, which implied H39_1 contributed larger genetic gains to ZD309 with regard to heat tolerance.

To interpret heat tolerance-associated molecular pathways in ZD309, all DEGs were subjected to GO and KEGG enrichment analyses (Figure 2C,D and Figure 3). For GO analysis, in both D3HS and D6HS treatments, the common DEGs were strongly enriched in photosynthesis, photosystem I, photosystem II, plasma membrane, chlorophyll-binding, chloroplast thylakoid membrane, and protein-chromophore linkage terms. The unique DEGs in D3HS treatment were in protein binding, mitochondrion, zinc ion binding, RNA modification, and response to light stimulus. The unique DEGs in D6HS treatment were in carbohydrate metabolic process, plastid, thylakoid, response to heat, carbohydrate transport. For KEGG analysis, the DEGs for D3HS treatment were mainly involved in photosynthesis-antenna proteins, cysteine and methionine metabolism, ABC transporters, starch and sucrose metabolism, and protein processing in endoplasmic reticulum pathways. The DEGs for D6HS treatment were mainly involved in photosynthesis, benzoxazinoid biosynthesis, carbon fixation in photosynthetic organisms, glycolysis/gluconeogenesis, and protein processing in endoplasmic reticulum pathways. These results illustrated that heat stress has strong effects on photosynthesis, protein processing, and starch and sucrose metabolism.

The overlapped DEGs among ZD309 and its parental lines in both D3HS and D6HS treatments were centered for further analysis (Figure 4A). A total of 299 DEGs (198 up- and 93 down-regulated) were suggested to be important for heat tolerance. These DEGs were enriched in photosynthesis-antenna proteins, protein processing in the endoplasmic reticulum, cyanoamino acid metabolism, and phagosome pathways (Figure 4B). Based on the gene expression patterns, these genes were classified into four groups, continued up-regulated genes, continued down-regulated genes, prolonged heat stress suppressed up-regulated genes, and prolonged heat stress mitigated up-regulated genes (Figure 4C). To correlate the RNA-seq results, the expression level of 10 randomly selected DEGs were analyzed by qRT-PCR (Figure 5).

Among the 198 up-regulated DEGs, 102 genes exhibited over-dominant effects (a higher expression level in ZD309 than in both parental lines), which is possible to explain the heat tolerance heterosis in the hybrid. Of these over-dominant effect genes, genes encoding for chaperone protein *ClpB1*, NADP-dependent oxidoreductase encoding gene, and *Hsp90* were enhanced in long time heat stress. Genes encoding for NAC transcription factor and zinc finger family proteins, *Zm00001d002905* (encode a protein kinase) and *Zm00001d048710* (encode benzoxazinone synthesis 2) were also enhanced in the hybrid. In addition, genes encoding for lysine histidine transporter, triose phosphate isomerase, protein kinases, and abscisic acid-inducible protein were activated by long-term heat stress. These highly up-regulated genes are involved in multiple biological processes, indicating that heat tolerance heterosis is associated with sophisticated regulatory networks.

In D3HS or D6HS treatments, 5504 (3463 up- and 2151 down-regulated) DEGs in hybrid ZD309 were identified. A total of 7324 (5189 up- and 2237 down-regulated) and 4505 (2538 up- and 2046 down-regulated) DEGs were identified in inbred lines H39_1 and M189 (Figure 6A). Of these DEGs, 1982 DEGs were solely expressed in the three tested materials, including 1171 up-regulated and 684 down-regulated genes (Figure 6B). The unique DEGs were subjected to KEGG pathway enrichment (Figure 6C). The up-regulated DEGs in ZD309 and its parental lines were strongly enriched in alanine and aspartate and glutamate metabolism, benzoxazinoid biosynthesis, cysteine and methionine metabolism, and ribosome biogenesis in eukaryotes. The up-regulated DEGs in H39_1 were preferably enriched in alpha-linolenic acid metabolism, amino sugar and nucleotide sugar metabolism, valine and leucine and isoleucine biosynthesis, amino sugar and nucleotide sugar metabolism, flavonoid biosynthesis, phenylalanine and tyrosine and tryptophan biosynthesis, and pentose phosphate. The up-regulated DEGs in M189 were preferably enriched in ABC transporters and starch and sucrose metabolism. Indeed, the up-regulated DEGs in ZD309 were preferably enriched in alpha-linolenic acid metabolism, carbon fixation in photosynthetic organisms, glycolysis/gluconeogenesis, linoleic acid metabolism, purine metabolism, phenylalanine and tyrosine and typtophan biosynthesis, fructose and mannose metabolism, and tyrosine metabolism. On the other hand, the down-regulated DEGs in ZD309 and its parental lines were greatly enriched in either photosynthesis or photosynthesis-antenna proteins. In comparison to the parental lines, the down-regulated DEGs in ZD309 were mainly enriched in brassinosteroid biosynthesis, limonene, and pinene degradation. Overall, genes associated with stress response (such as response to biotic stimulus, response to water deprivation, and response to hydrogen peroxide) were preferably enriched in up-regulated DEGs in ZD309. DEGs in inbred line H39_1 share more common pathways with ZD309 than inbred line M189.

### 2.4. The Expression of HSFs and HSPs Genes in the DEGs

Of the detected DEGs, 15 *hsf-tfs* (HSFs encoding genes) were differentially expressed under heat treatments (Table 2). In D3HS treatment, most *hsf-tfs* were down-regulated or insignificant-expressed. However, *hsf-tf18* was enhanced in both ZD309 and H39_1; *hsf-tf6* was elevated in M189; *hsf-tf17* and *hsf-tf9* were up-regulated specifically in H39_1; and *hsf-tf7* did solely in ZD309. Interestingly, *hsf-tf13* was down-regulated in the three tested materials. Compared with the D3HS treatment, more *hsf-tfs* were enhanced in D6HS treatment, suggesting that prolonged heat stress triggers more *hsf-tfs* to be differently expressed. *hsf-tf18* and *hsf-tf7* were up-regulated in ZD309 and its parental lines. *hsf-tf20*, *hsf-tf24* and *hsf-tf28* were up-regulated in ZD309. Most *hsf-tfs* were up-regulated in ZD309, followed by H39_1, then the least for M189.

Of the detected DEGs, 15 *hsp* (HSPs encoding genes) were differentially expressed under heat treatments (Table 2). In D3HS and D6HS treatments, *traf21* (encodes TNF receptor-associated factor 21, a heat-response protein), *hsp13*, *Hsp20* were down-regulated in all three tested materials. However, *hsp19*, *Hsp20*, *Zm00001d015777* (encodes a chloroplast small heat shock protein), *B6SIX0* (encodes 16.9 kDa class I heat shock protein 1), *hsp26*, and *hsp11* were up-regulated in ZD309 and its parental lines. Strikingly, *hsp10*, *hsp22*, *mbf2* (encodes multiprotein-bridging factor 1c), *gpc3* (encodes glyceraldehyde-3-phosphate dehydrogenase3) were up-regulated in ZD309 and H39_1 in two heat treatments but down-regulated in M189 in D3HS treatment. Overall, more *hsps* and heat response-associated genes were activated in the D6HS treatment than in the D3HS treatment.

### 2.5. Metabolic Analysis of in ZD309 and Its Parental Lines Response to Heat Stress

To examine the metabolic response to heat stress in ZD309 and its parental lines, leaf samples were collected in D3HS and D6HS treatments for untargeted metabolite analysis based on ultra-performance liquid chromatography/mass spectrometry (UPLC/MS) platform. PLS-DA (partial least squares discriminant analysis) was performed to identify the heat-response metabolites through differential abundance. The differentially expressed metabolites (DEMs) were defined as a criterion of *p*-value ≤ 0.05, |log2FC (fold changes)| ≥ 1, and VIP (variable importance in projection) score ≥ 1. The metabolite data have been deposited into the CNGB Sequence Archive (CNSA) of China National GeneBank DataBase (CNGBdb) with accession number CNP0002688. In D3HS treatment, 1542, 1150, and 1401 DEMs were identified in the heat-stressed H39_1, M189, ZD309 versus the corresponding controls, respectively (Figure 7A). Of these DEMs, 754 DEMs were shared between H39_1 and ZD309, 396 DEMs shared between M189 and ZD309, as well as 357 DEMs between H39_1 and M189. A total of 201 overlapped DEMs were found among the three tested materials. In D6HS treatment, 1782, 1801, and 2374 DEMs were identified in the heat-stressed H39_1, M189CK, ZD309CK versus the corresponding controls, respectively (Figure 7B). Of these DEMs, 1295 DEMs were shared between H39_1 and ZD309, 1102 DEMs shared between M189 and ZD309, as well as 946 DEMs between H39_1 and M189. A total of 747 overlapped DEMs were found among the three tested materials (Figure 7B).

Of these detected DEMs, 350, 311, and 423 DEMs were up-regulated in H39_1, M189, and ZD309, respectively (Figure 7C). A total of 150 DEMs were found across the tested materials, which might be essential for heat tolerance. Under heat stress, some metabolites were enhanced, including sugars and glycosides (e.g., mannose 6-phosphate, isopentenyladenine-9-N-glucoside, and cytarabine), methionine and organic acids (e.g., jasmonic acid (M211T234), dihydrojasmonic acid, and 9-Oxo-10(E), 12(E)-octadecadienoic acid), nucleosides and nucleotides (e.g., cytosine and 5-Methylcytosine) (Figure 7D, Table 3). In contrast, other metabolites, amino acids (e.g., lysine, isoleucine, and aspartic acid), and alkaloids were reduced (Figure 7D, Table 3). Furthermore, 41, 71, and 90 metabolites were found to be associated with heat stress specifically in H39_1, M189, and ZD309, respectively. After heat treatment, JA abundance was significantly increased in three tested materials versus the corresponding controls. JA abundance was also significantly increased in D6HS treatment compared with that in D3HS treatment.

### 2.6. Integrative Analysis of Gene Expression and Metabolic Changes under Heat Stress Conditions

To decipher the transcriptional regulation of metabolic networks, a co-enriched transcriptome and metabolome pathway analysis was conducted. In D3HS treatment, 38, 26, 26 co-enriched pathways were found in H39_1, M189, and ZD309, including starch and sucrose metabolism, arginine and proline metabolism, isoquinoline alkaloid biosynthesis, and glutathione metabolism. In D6HS treatment, 36, 22, and 37 co-enriched were found in H39_1, M189, and ZD309, including carbon fixation in photosynthetic organisms, phenylpropanoid biosynthesis, alpha-linolenic acid metabolism, and glycerophospholipid metabolism. Importantly, six co-enriched pathways were detected in both heat stress treatments, including plant hormone signal transduction, cysteine and methionine metabolism, alanine and aspartate and glutamate metabolism, ABC transporters, zeatin biosynthesis, and cyanoamino acid metabolism. These pathways might play critical roles in conferring maize heat tolerance.

Most DEGs were also identified in these co-enriched pathways. The expressions of these DEGs were highly consistent with the contents of corresponding metabolites. For instance, mannose-6-phosphate, an intermediate of fructose and mannose metabolism, was elevated among the three tested materials. Most fructose and mannose metabolism-related genes were activated under heat stress, with a remarkable level in the inbred line H39_1 (Figure 8).

### 2.7. The Expression of Genes Involved in JA Biosynthesis under Heat Stress

Combined transcriptome and metabolome analyses have demonstrated that JA is a key regulator to confer heat tolerance. Upon metabolomics, JA raised significantly under heat stress. Compared with M189, the JA contents (M211T234) were increased in H39_1 and ZD309 (Figure 9A). The JA levels were enhanced along with the heat stress. The JA contents (M211T234) in D3HS treated H39_1, M189, ZD309 were increased 4.8, 2.5, and 4.6 folds versus the corresponding controls, respectively. The JA (M211T234) contents in D6HS treated H39_1, M189, ZD309 were increased 21.7, 27.9, and 73.3 folds versus the corresponding controls, respectively. The expression pattern of JA biosynthesis (α-linolenic acid metabolism) genes under heat stress was further analyzed (Figure 9B). Overall, most genes were up-regulated among three materials, particularly in H39_1 and ZD309. *Zm00001d053675* (encoding LOX) were up-regulated among the three tested materials, with the highest level in D3HS-treated ZD309 plants. However, six maize AOS encoding genes showed a higher level in H39_1 and ZD309 materials of D3HS treatment. Two AOC encoding genes, *Zm00001d029594* and *Zm00001d047340*, were up-regulated among the three tested materials under heat stress, with a relatively lower level of *Zm00001d029594*. Two OPCL1 encoding genes, *Zm00001d027519* and *Zm00001d043376*, were also up-regulated among the tested materials. In contrast, *Zm00001d043376* showed a higher level in D6HS than D3HS treatments. Moreover, *Zm00001d035854*, encoding α-dioxygenase, was the least level in ZD309 in D6HS treatment.

## 3. Discussion

### 3.1. ZD309 Is a Heat-Tolerant Maize Hybrid

As a sessile organism, maize suffers from several climate stresses, such as salinity, drought, heat, lodging, and flooding. Heat stress particularly hinders maize grain yield in the Huang-Huai-Hai River Basin area of China. In plants, heat stress generates excess reactive oxygen species (ROS) in the cellular compartments, such as chloroplasts, mitochondria, plasma membranes, peroxisomes, apoplast, and ER [2]. Excessive ROS alters a series of physiological responses, including photosynthesis, respiration, transpiration, membrane thermostability, and osmotic regulation [3]. Finally, the development and productivity of plants are largely inhibited [3]. In the present study, ZD309 was screened for less yield loss under heat stress and demonstrated superior heat tolerance. Further heat treatments on the ZD309 and its parental lines seedlings resulted in multiple morphological and physiological alterations. Consistent with the previous studies [1,3,23,24,25,26,27], heat stress interrupted plant growth and yielded increased levels of proline, MDA, H2O2, soluble sugar, soluble protein, SOD, POD, and CAT, as well as decreased level of chlorophyll a contents. Although ZD309 and its parental lines indicated superior heat tolerance, they had different mechanisms to minimize heat stress. The plant height of ZD309 and H39_1 were more affected by heat stress, but they had relatively stronger growth vigor. The growth of M189 was less affected, but it produced a relatively higher content of ROS content and low levels of SOD, POD, and CAT activities. These results indicated that M189 has a stronger heat tolerance compared with ZD309 and H39_1. ZD309 and H39_1 likely minimized the effects of heat stress through growth adaptation.

### 3.2. Key HSFs and HSPs Identified to Mediate Maize Heat Tolerance

In plants, heat stress triggers protective mechanisms to mitigate the damage from heat stress. Heat shock factors (HSFs) and heat shock proteins (HSPs) play crucial roles in such protective mechanisms [2,3]. Unlike mammalian cells, most plant HSFs are self-regulated by heat shock with some up-regulated, whereas others down-regulated. In *Arabidopsis*, the regulation of HSFs forms a transcriptional cascade in which HSFs activate each other to response environmental stress [2,28]. In *Arabidopsis* and maize, 21 and 31 HSFs were identified, respectively [2,29]. Extensive studies have demonstrated that HSFA2 is a central regulator in this cascade, while HSFA1s are responsible for heat shock response [2,3,11,30]. In maize, several HSFs were up-regulated in heat-treated maize seedlings (at V4 and V5 stage), such as *Zm00001d027757* (*hsftf13*, type A HSF), *Zm00001d032923* (*hsftf24*, A2 subgroup HSF), *Zm00001d046204* (*hsftf12*, A6 subgroup HSF), *Zm00001d052738* (*hsftf7*, B2 subgroup HSF), *Zm00001d026094* (*hsftf20*, B2 subgroup HSF), *Zm00001d016674* (*hsftf6*, A3 subgroup HSF), *Zm00001d005888* (*hsftf1*, B1 subgroup HSF), *Zm00001d016255* (*hsftf18*, C2 subgroup HSF), and *Zm00001d046299* (*hsftf28*, C2 subgroup HSF). However, *Zm00001d020714* (*hsftf21*, B1 subgroup HSF) was down-regulated [2,29,31,32]. In the present study, *Zm00001d016255* (*hsftf18*), *Zm00001d016674* (*hsftf6*), and *Zm00001d052738* (*hsftf7*) were up-regulated; *Zm00001d018941* (*hsftf4*), *Zm00001d026094* (*hsftf20*), *Zm00001d027757* (*hsftf13*), *Zm00001d020714* (*hsftf21*), *Zm00001d033987* (*hsftf17*), and *Zm00001d038746* (*hsftf15*) were down-regulated. These differentially expressed HSFs are likely to function in ZD309 and its parental lines under heat stress. Notably, the expression pattern of most HSFs were similar between ZD309 and H39_1 but varied in M189.

In response to heat and other stresses, HSFs cascade subsequently activate the expression of HSPs [10,33]. In plants, HSPs are grouped into five classes regarding to molecular weight (kDa): HSP100, HSP90, HSP70, HSP60, and small heat shock proteins (sHSPs) [33,34,35]. Generally, HSPs are thought to function as chaperones to alleviate protein misfolding and aggregation problems. In maize, similar to *Arabidopsis*, heat shock activates the expression of both high and low molecular weight HSPs [36,37,38]. In the present study, we identified several HSPs that were up-regulated under heat stress, such as *Zm00001d048073* (*hsp19*), *Zm00001d024903* (*hsp90*), *Zm00001d028408* (*hsp26*), *Zm00001d028561* (*hsp11*), *Zm00001d028555* (*hsp10*), *Zm00001d052194* (*hsp22*), *Zm00001d044728* (*hsp27*), and *Zm00001d015777* (encodes a chloroplast small heat shock protein), and *B6SIX0* (encodes 16.9 kDa class I heat shock protein 1).

### 3.3. Important Metabolites in Response to Heat Stress

Abiotic stresses can also cause metabolic alterations in plants, such as maintaining essential metabolism and synthesizing protective metabolites to protect themselves from stress damage. Heat stress is a major abiotic stress in crop production. By untargeted metabolomics analysis, metabolic reprogramming was identified in several crops, such as tomato, maize, barley, wheat, and soybean [16,21,39,40,41]. Hundreds of metabolites were indeed identified that are closely associated with heat stress, such as malate, valine, isoleucine, glucose, starch, sucrose, proline, glycine, serine, drummondol, anthranilate, dimethylmaleate, galactoglycerol, guanine, and glycerone. Heat stress also caused endogenous hormone changes, including zeatin, gibberellin acid3 (GA3), gibberellin acid4 (GA4), abscisic acid (ABA), salicylic acid (SA), and jasmonic acid (JA) [15]. In the present study, 150 DEMs were identified in the tested three materials. Of these DEMs, mannose 6-phosphate, isopentenyladenine-9-N-glucoside, cytarabine, JA, octadecadienoic acid, cytosine, and 5-methylcytosine were highly accumulated under heat stress. Conversely, lysine, isoleucine, aspartic acid, alkaloids were significantly reduced. These enhanced metabolites might contribute to the heat tolerance of ZD309 and its parental lines. Most DEMs were correlated with DEGs, which mainly co-enriched in several GO terms, including plant hormone signal transduction, cysteine, and methionine metabolism, alanine and aspartate and glutamate metabolism, ABC transporters, zeatin biosynthesis, and cyanoamino acid metabolism. The key metabolites can be used to enhance maize heat tolerance through exogenous metabolites feeding.

### 3.4. Potential Applications of HSFs, HSPs, and Metabolites in Heat Stress Tolerance Improvement

With an increasing number of essential heat-response genes characterized, the underlying regulatory network has been partly resolved in model plants. However, there is still a gap for heat resilient variety breeding in crop plants. In the present study, some major HSFs, HSPs, and metabolites were identified through combined transcriptomic and metabolomic analysis. Of them, these identified HSFs, HSPs genes can be used for screening ideal haplotypes, thereby selecting heat-tolerant inbred lines and creating heat-tolerant hybrid varieties. In previous studies, the SA187 [42] and biochar [43] application enhanced heat tolerance in *Arabidopsis* and rice, respectively. Similarly, the identified metabolites can be used for improving maize tolerance by exogenous application. Taken together, our results will not only be a benefit for uncovering the mechanism of the heat-response network but also can provide the information for heat tolerance improvement.

## 4. Materials and Methods

### 4.1. Plant Materials and Heat Treatment

A total of 12 commercial maize hybrids, including ZD309 (Henan Academy of Agricultural Sciences, China, dent grain, summer maize), were potted planted at Yuanyang Farm of Henan Academy of Agricultural Sciences. The experimental soil belongs to yellow-cinnamon soil, which contains total N for 0.74 g kg^−1^, available P for 42.6 mg kg^−1^, 163.0 mg kg^−1^, and organic matter for 33.0 g kg^−1^ (pH 8.83). To estimate heat stress effects on different maize hybrids, 50 six-week-old plants from each hybrid were selected under consistent growth conditions. The selected plants were divided into two groups, with 25 plants each. One group was moved to a sunlight greenhouse for heat stress treatment, and the other group was placed in the field as the control. In the sunlight greenhouse, the daytime setting temperature was 35 °C (actual temperature 35.0–43.6 °C, outdoor air temperature 32.5–38.7 °C), the nighttime setting temperature was 28 °C (actual temperature 28.0–33.7 °C, outdoor air temperature 24.5–30.4 °C). On average, the temperature in the greenhouse was 3–5 °C higher than the field. All plants were well watered every two days to ensure no water stress. At the kernel maturity stage, the ears from the two groups were harvested for estimating the grain yield. The potted experiment was repeated three times.

The hybrid ZD309 and its parental lines, H39_1 and M189, were germinated and grown in the greenhouse at 28/24 °C (12/12 h dark/light cycle) with 60% relative humidity and 1000 µmol m^−2^ s^−1^ light density. The uniform of five-week-old maize seedlings were selected and divided into two groups and three replications, with 24 seedlings for each group and 8 seedlings for each replication. One group was moved into a growth chamber with a 28/24 °C (day/night) cycle as the control (CK). The other group was moved into a growth chamber with a 40/35 °C(day/night) cycle as the heat treatment (HT) stress. The plants were well watered every two days to ensure no water stress. The uppermost whole leaves of maize seedlings were sampled after heat treatment for 3 days (D3HS) and 6 days (D6HS), frozen in liquid nitrogen, and stored at −80 °C for the following experiments. Each sample had three biological replicates.

### 4.2. Heat Stress-Related Physiological Parameters Measurement

The leaf samples of ZD309 and its parental lines were subjected to physiological parameter determination. The contents of activity of proline, malondialdehyde (MDA), H_2_O_2_, soluble sugar, soluble protein, chlorophyll, superoxide dismutase (SOD), peroxidase (POD), and catalase (CAT) were measured using the corresponding testing kit from Suzhou Comin Biotechnology Co., Ltd. (Suzhou, China. Available online: http://www.cominbio.com/a/shijihe/, accessed on 1 January 2022) by following the manufacturer’s instructions. These testing kits include PRO-2-Y, MDA-2-Y, H2O2-2-Y, KT-2-Y, BCAP-2-W, Cha-4-H, Chb-4-H, SOD-2-W, POD-2-Y, and CAT-2-W.

### 4.3. mRNA Library Construction and Illumina Sequencing

Total RNA was extracted from maize leaves of the three materials using Trizol reagent (Invitrogen, CA, USA) according to the manufacturer’s instructions. The total RNA quality was evaluated by Bioanalyzer 2100 and RNA 1000 Nano LabChip Kit (Agilent, CA, USA) with a RIN number > 7.0. Poly(A) RNA was first purified from total RNA (5 ug) with poly-T-oligo-attached magnetic beads for two rounds of purification. The full-length mRNA was next fragmented into small pieces using divalent cations under elevated temperature. The cleaved RNA fragments were then reverse transcribed into cDNA for library construction by using the mRNA-Seq sample preparation kit (Illumina, San Diego, CA, USA). The average insert size for the paired-end libraries was 300 bp (±50 bp). Paired-end sequencing was performed on the Illumina Hiseq 4000 platform (LC Sciences, Houston, TX, USA).

### 4.4. Transcriptome Analysis

Clean reads were aligned to the B73 maize reference genome (B73 RefGen_V4.42, available online: http://ensembl.gramene.org/Zea_mays/Info/Index, accessed on 1 January 2022) using the HISAT2 V2.1.0 [44]. The mapped reads for each sample were assembled using StringTie (available online: https://ccb.jhu.edu/software/stringtie/, accessed on 1 January 2022) [45]. All subset transcriptomes from each sample were merged to build a comprehensive transcriptome using perl scripts. When the whole transcriptome was done, StringTie and edgeR (available online: http://bioconductor.org, accessed on 1 January 2022) [46] was used to estimate the expression levels of all transcripts. The differentially expressed genes (DEGs) were filtered with |log_2_ (fold change)| > 1 and statistical significance (*p*-value < 0.05) by R package. Gene Ontology (GO) enrichment was performed by using agriGO (available online: http://bioinfo.cau.edu.cn/agriGO/index.php, accessed on 1 January 2022) [47]. GO terms with FDR ≤ 0.05 were significantly enriched. Kyoto Encyclopedia of Genes and Genomes (KEGG) enrichment was performed by using an online server (www.genome.jp/kegg, accessed on 1 January 2022) [48]. KEGG pathways with FDR ≤ 0.05 were significantly enriched.

### 4.5. Quantitative Real-Time PCR

To verify the reproducibility of DEGs in the RNA-seq data, we used the same samples for evaluating the gene expression levels by qRT-PCR. Ten genes from the DEGs were selected randomly, and corresponding primers were designed by NCBI (Primer-Blast). The gene name and primer sequence of the DEGs are listed in Appendix A. Total RNA of the tested samples in RNA-seq reverse transcribed using the PrimeScript™ RT reagent kit with gDNA Eraser (Perfect Real Time) (Takara, Dalian, China). Then, qRT-PCR was performed by using the Gene Applied Biosystems@7500 Fast and TransStart Top Green qPCR SuperMix (TransGen Biotech, Beijing, China). Each sample was analyzed with three technical replicates. The PCR cycle was configured as: 5 min at 95 °C followed by 40 cycles of amplification at 95 °C for 15 s, then 58 °C for 20 s, and 72 °C for 30 s. The relative expression level of the DEGs was calculated by the 2^−ΔΔCt^ [49]. The GAPDH gene was used as the internal control [50].

### 4.6. Metabolites and Metabolome Analysis

The samples were placed on ice, and the metabolites were extracted with 50% methanol buffer [51]. Briefly, 100 mg of sample was ground into powder and added 120 μL of precooled 50% methanol, vortexed for 1 min, and incubated at room temperature for 10 min; the extraction mixture was then stored overnight at −20 °C. After centrifugation at 4000× *g* for 20 min, the supernatants were transferred into new 96-well plates and stored at −80 °C refrigerator for further UPLC-MS analysis. In addition, pooled QC samples were also prepared by combining 10 μL of each extraction mixture. A quality control (QC) sample was included to check the repeatability within an analytical batch.

The MS data were analyzed using XCMS and CAMERA software (available online: https://bioconductor.org/, accessed on 1 January 2022) [52]. To obtain high-quality data, robust Loess signal correction (R-LSC) was used to normalize peaks from the tested samples with regard to the QC samples, and the relative standard deviation (RSD) was calculated. Those features that were detected in less than 50% of QC samples or 80% of biological samples were removed, then the remaining peaks with missing values were imputed with the k-nearest neighbor algorithm to further improve the data quality. The *m*/*z* with RSD value between 0% and 30% was subjected to principal component analysis (PCA), partial least squares discriminant analysis (PLS-DA), and variable importance in the projection (VIP) analysis. The *m*/*z* with VIP  >  1, fold change > 1.5 (or < 0.67) and Q value  <  0.05 in any sample was retained for further metabolite identification (at identification level 2) and pathway annotation by using the online HMDB database (available online: http://www.hmdb.ca/, accessed on 1 January 2022), METLIN (available online: http://metlin.scripps.edu/, accessed on 1 January 2022) and KEGG (www.genome.jp/kegg/, accessed on 1 January 2022) [53,54].

### 4.7. Statistical Analysis

All data collected from phenotypic, physiological parameters and qRT-PCR analyses were subjected to one-way variance analysis (ANOVA) and Student’s *t*-test using software SPSS 22.0 (IBM, New York, NY, USA). *p* < 0.05 indicates the statistical differences to reach the significant different level, *p* < 0.01 for very significant different levels.

## Figures and Tables

**Figure 1 plants-11-00677-f001:**
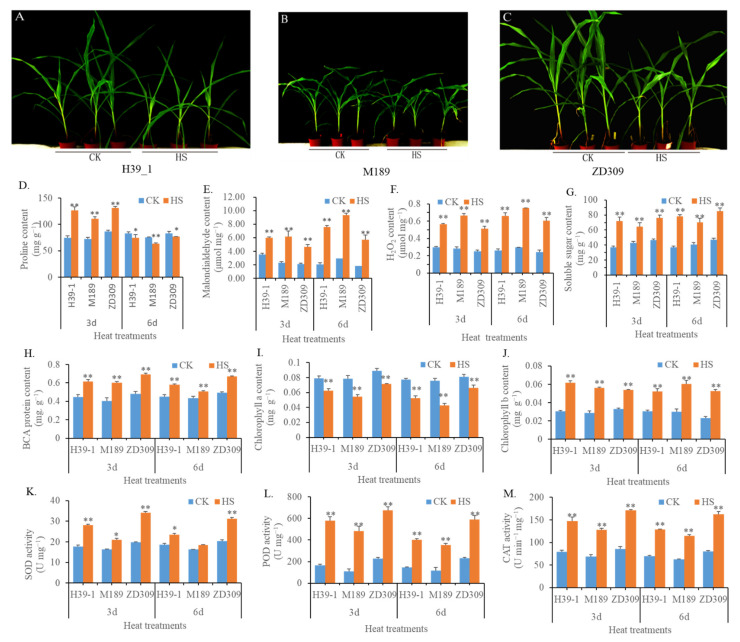
The effects of heat stress on maize growth and physical characters in ZD309 and its parental lines. (**A**–**C**) Performance of maize plants treated by 6 days heat stress in H39_1 ((**A**), maternal line), M189 ((**B**), paternal line), and ZD309 (**C**); (**D**–**M**) Physical alterations mediated by heat treatments, including proline content (**D**), malondialdehyde content (**E**), H_2_O_2_ (**F**), soluble sugar (**G**), BCA protein (**H**), chlorophyll a/b content (**I**,**J**), as well as SOD, POD, and CAT activity. For the content of proline, soluble sugar content, BCA protein, chlorophyll a/b, the measurement unit, mg g^−1^, represents the respective content in 1 g of sample’s fresh weight; For the content of malondialdehyde and H_2_O_2_, the measurement unit, μmol mg^−1^, represents the respective content in 1 mg of sample’s fresh weight; For the activity of SOD and POD, the measurement unit, U mg^−1^, represents the respective activity units detected in 1 mg of sample’s fresh weight; For CAT activity, the measurement unit, U min^−1^ mg^−1^, 1 activity unit (U) represents the A240 absorbance decreased 0.1 OD in 1 min and 1 mg of sample’s fresh weight. * and ** represent that the corresponding physical character in heat-stressed maize plants are significantly and very significantly different from the control at *p* < 0.05 and *p* < 0.01 levels, respectively.

**Figure 2 plants-11-00677-f002:**
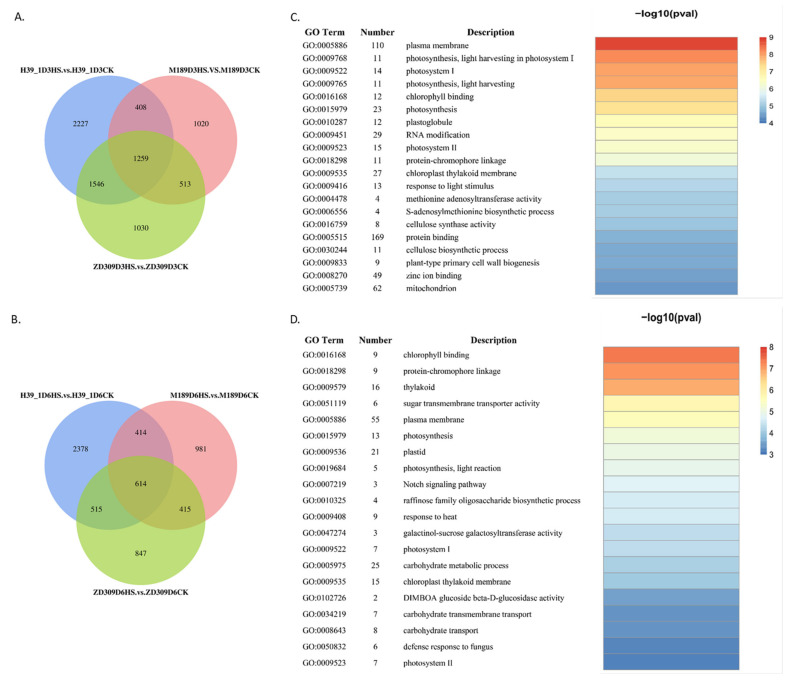
DEGs identifications and gene ontology analysis in transcriptomic data. (**A**,**B**) Numbers of the identified DEGs in the tested three genotypes under D3HS and D6HS treatments. (**C**,**D**) Gene ontology analysis for the conserved DEGs within ZD309 and its parental lines in H3HS (**C**) and H6HS (**D**) treatments. The heatmap presents statistical significance (log10-transformed corrected *p*-value) of GO terms over-representation.

**Figure 3 plants-11-00677-f003:**
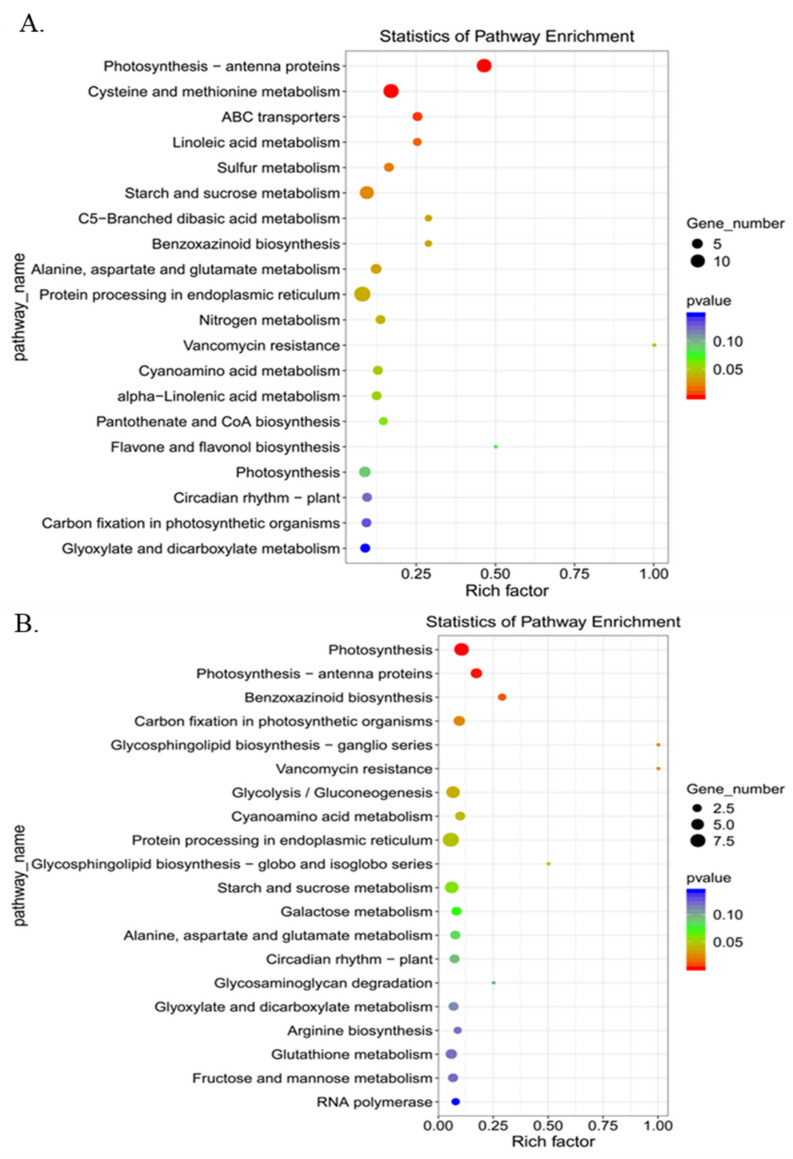
Kyoto Encyclopedia of Genes and Genomes (KEGG) enrichment analysis for the conserved DEGs within ZD309 and its parental lines in H3HS (**A**) and H6HS (**B**) treatments.

**Figure 4 plants-11-00677-f004:**
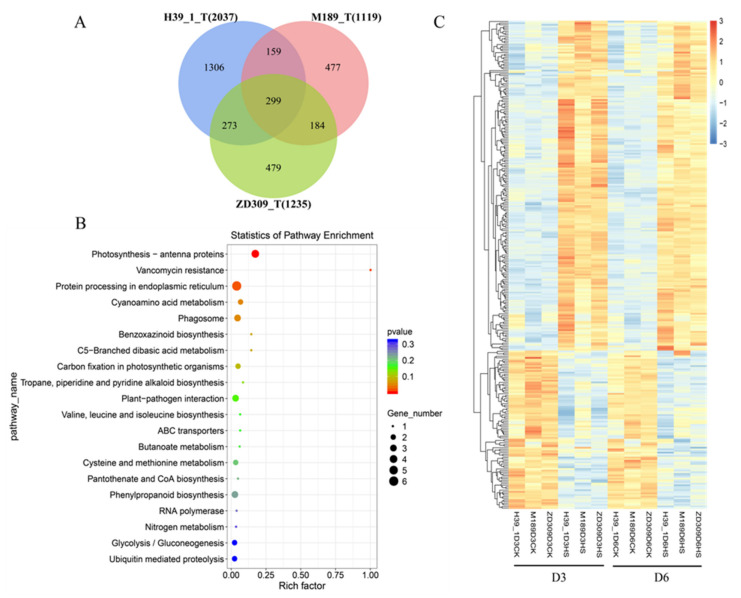
The conserved DEGs identified between the two heat treatments. (**A**) Numbers of DEGs between the two heat treatments in the tested three genotypes; (**B**) Kyoto Encyclopedia of Genes and Genomes (KEGG) enrichment analysis for between the two heat treatments in the tested three genotypes. The x-axis indicated the rich factor, and the y-axis indicated pathway; (**C**) expression patterns of the identified 299 conserved DEGs. The color scale at the right of the figure indicates the gene expression levels transformed by log_10_ (FPKM + 1).

**Figure 5 plants-11-00677-f005:**
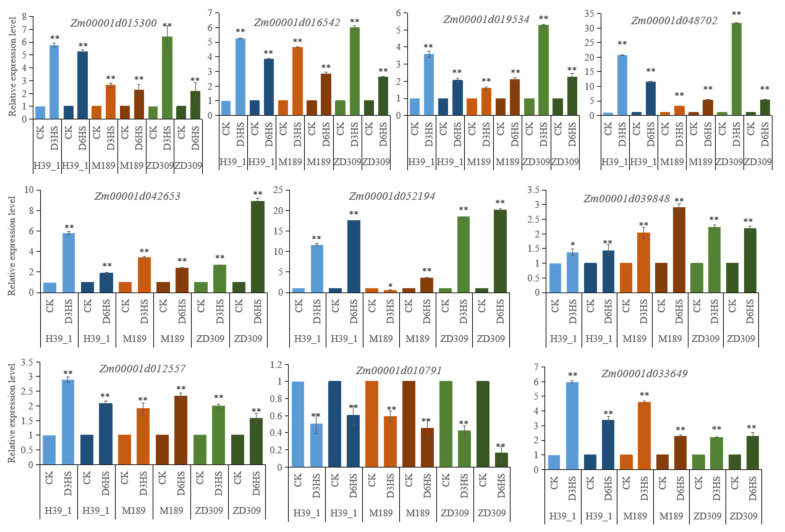
Verification the expression of DEGs by qRT-PCR. Bars show standard error of the relative expression levels. * and ** represent that the corresponding gene expression level in heat-stressed maize plants are significantly and very significantly different from the control at *p* < 0.05 and *p* < 0.01 levels, respectively.

**Figure 6 plants-11-00677-f006:**
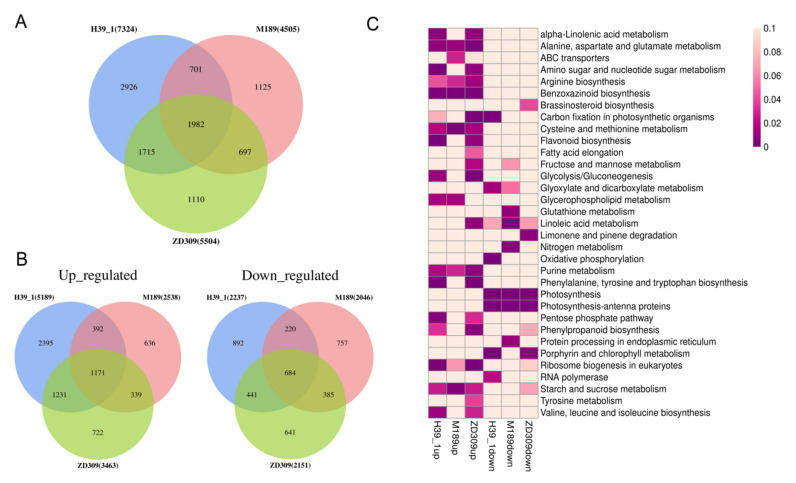
Comparison of the identified DEGs in ZD309 and its parental lines. (**A**) Numbers of DEGs of ZD309 and its parental lines in D3HS or D6HS treatment. (**B**) Numbers of up- and down-regulated DEGs of ZD309 and its parental lines in D3HS or D6HS treatment. (**C**) KEGG enrichment analysis for the up- and down-regulated DEGs in each genotype. The heatmap presents statistical significance (*p*-value) of KEGG pathway term over-representation.

**Figure 7 plants-11-00677-f007:**
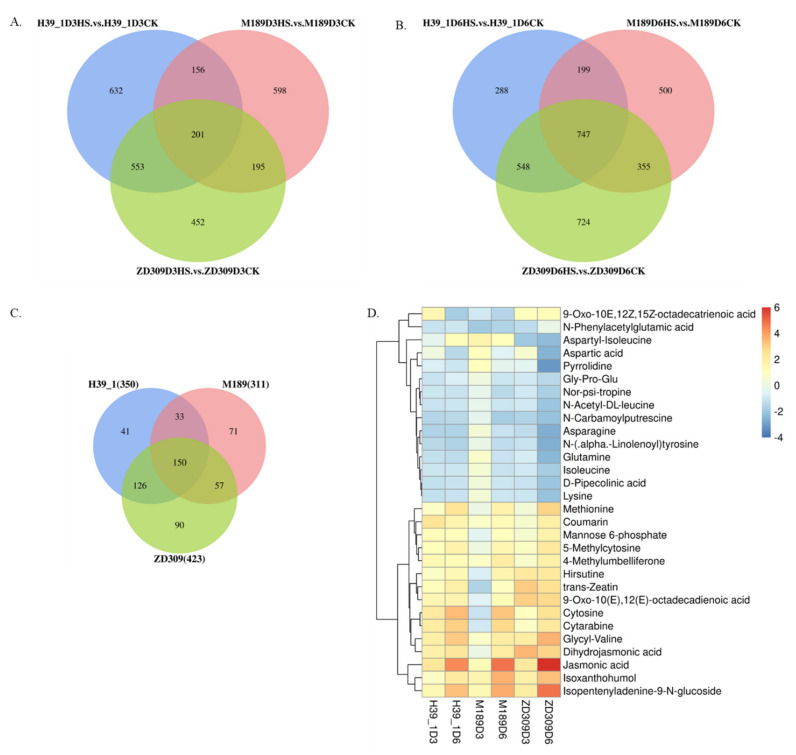
DEMs and expression pattern identifications within ZD309 and its parental lines. (**A**) DEMs numbers of ZD309 and its parental lines in D3HS treatment; (**B**) DEMs numbers of ZD309 and its parental lines in D6HS treatment. (**C**) Conserved DEGs numbers of ZD309 and its parental lines between D3HS and D6HS. (**D**) Expression patterns of the conserved DEMs between the two heat treatments.

**Figure 8 plants-11-00677-f008:**
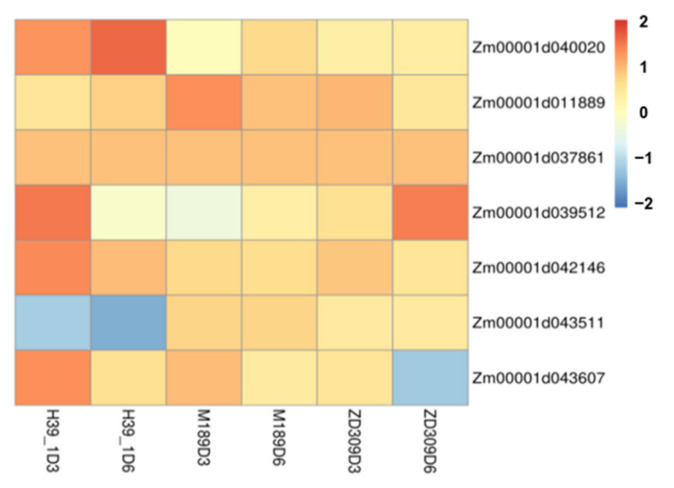
The relative expression levels of 7 fructose and mannose metabolism-related genes in the tested three plant materials. Data are presented as heatmaps of a log_2_ transformed fold change.

**Figure 9 plants-11-00677-f009:**
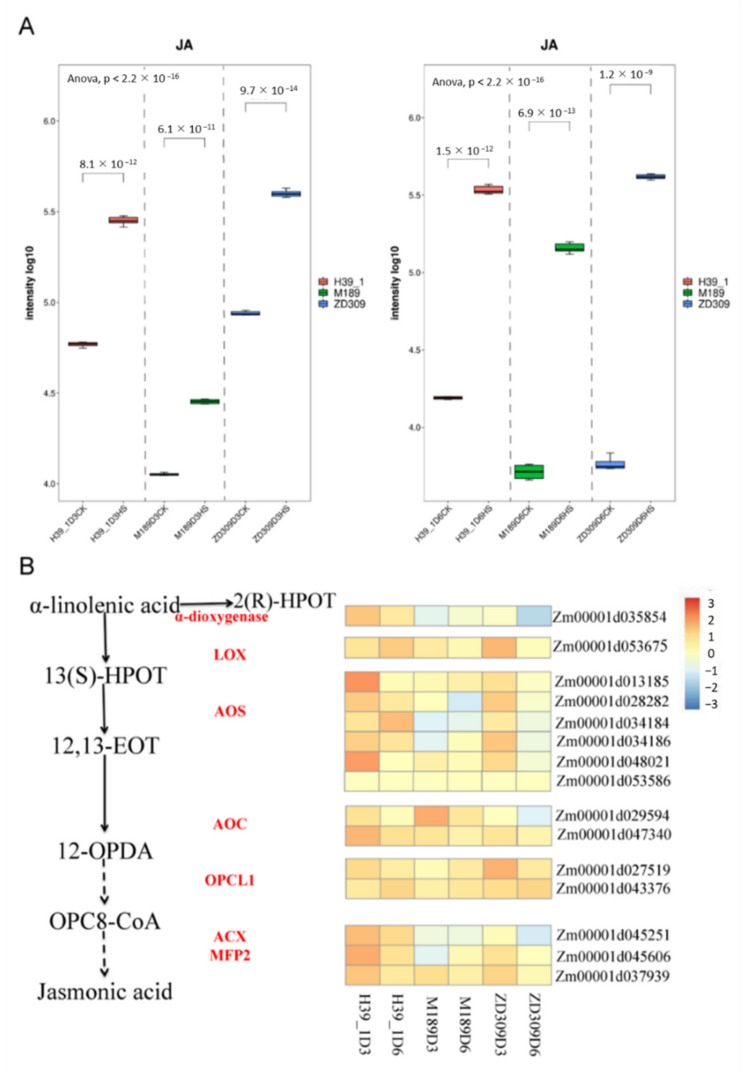
The alterations of JA levels and metabolism pathway-related genes expression levels in response to heat stress. (**A**) JA levels in three genotypes at both heat treatments; (**B**) The relative expression of the genes involved in JA biosynthesis. Data are presented as a heatmap by a log_2_ transformed fold change.

**Table 1 plants-11-00677-t001:** Identification of the effects of heat stress on grain yield in 12 commercial hybrids.

Varieties	CK		HS Treated		Heat-Tolerant Coefficient (%)
	200-Kernel Weight (g)	Grain Yield (kg/ha)	200-Grains Weight (g)	Grain Yield (kg/ha)	
Zhengdan1002	66.53	8773.8	60.85	2362.35	26.92
Zhengdan326	64.07	8029.05	58.28	2773.65	34.54
Pionner335	69.12	5573.55	63.11	2142.75	38.44
Zhengdan958	60.38	7745.55	49.34	3033.45	39.16
Weike702	64.16	8480.7	47.48	3852	45.42
Denghai605	61.03	7574.55	54.91	4795.5	63.31
Nonghua101	71.92	6187.8	69.69	4560.6	73.7
Qiule368	72.31	8304.15	64.98	6133.65	73.86
Zhengdan309	60.73	8004.45	64.66	6591.6	82.35
DK653	66.6	5195.7	59.00	4371.6	84.14
Zhongkeyu505	65.26	4392.45	56.45	4584.3	104.37
Zhengdan317	67.27	6952.05	61.14	7366.2	105.96

**Table 2 plants-11-00677-t002:** The expression alterations of the HSF and HSP/heat-response proteins encoding genes under heat stress.

Gene_ID	Gene Name	D3HS			D6HS		
		H39_1VSCK	M189VSCK	ZD309VSCK	H39_1VSCK	M189VSCK	ZD309VSCK
*Zm00001d005888*	*hsf−tf1*	0.30	−1.37	−0.55	−0.10	−0.39	−0.91
*Zm00001d010812*	*hsf−tf16*	0.02	−1.25	−0.75	0.19	0.21	0.33
*Zm00001d016255*	*hsf−tf18*	1.46	0.19	3.49	3.45	1.52	3.92
*Zm00001d016674*	*hsf−tf6*	0.09	1.36	0.91	0.76	0.78	0.89
*Zm00001d018941*	*hsf−tf4*	−3.00	−2.09	−1.22	0.01	−2.13	−0.62
*Zm00001d020714*	*hsf−tf21*	0.63	−3.11	−0.03	0.23	0.27	−0.26
*Zm00001d021263*	*hsf−tf10*	−1.72	−3.25	−1.17	0.07	−0.56	0.27
*Zm00001d026094*	*hsf−tf20*	−3.54	−6.31	−0.93	−0.20	−2.26	1.35
*Zm00001d027757*	*hsf−tf13*	−4.54	−4.35	−2.10	−1.97	−3.48	−0.61
*Zm00001d032923*	*hsf−tf24*	−1.80	−1.11	0.03	0.57	−0.57	1.03
*Zm00001d033987*	*hsf−tf17*	1.12	−8.42	−1.76	−1.16	−1.28	0.63
*Zm00001d038746*	*hsf−tf15*	−0.67	−1.65	−0.86	−0.23	−0.19	−0.18
*Zm00001d046299*	*hsf−tf28*	−1.59	−1.65	0.09	1.21	−0.23	1.04
*Zm00001d048041*	*hsf−tf9*	1.19	0.65	−0.17	−0.02	−0.60	−1.32
*Zm00001d052738*	*hsf−tf7*	−0.33	−1.62	1.20	1.15	1.51	1.33
*Zm00001d039469*	*traf21*	−1.59	−1.98	−2.06	−0.95	−1.96	−1.97
*Zm00001d052855*	*hsp13*	−1.79	−3.92	−1.83	−0.43	−1.79	−0.70
*Zm00001d048073*	*hsp19*	3.06	2.24	2.37	1.53	0.93	1.68
*Zm00001d024903*	*hsp90*	7.27	3.79	7.64	5.02	4.49	6.30
*Zm00001d015777*	*Zm00001d015777*	3.57	2.15	14.58	4.01	2.02	2.08
*Zm00001d039933*	*B6SIX0*	4.28	2.66	3.42	1.97	1.97	2.29
*Zm00001d028408*	*hsp26*	7.55	2.15	2.51	5.17	3.31	3.16
*Zm00001d028561*	*hsp11*	8.30	4.97	5.01	3.55	3.42	7.23
*Zm00001d028555*	*hsp10*	1.72	−1.09	1.23	1.89	0.03	1.82
*Zm00001d052194*	*hsp22*	3.47	−1.09	4.78	3.81	1.33	4.44
*Zm00001d021634*	*Hsp20*	−0.91	−0.62	−1.45	−2.40	−1.56	−1.24
*Zm00001d050346*	*Annexin*	1.24	0.59	−0.10	1.19	1.85	1.17
*Zm00001d046471*	*mbf2*	2.80	−0.80	8.16	5.51	3.09	5.60
*Zm00001d044728*	*hsp27*	0.56	0.82	1.47	2.20	1.66	1.85
*Zm00001d051001*	*gpc3*	1.99	−0.37	2.10	3.58	1.37	2.21

Note: The expression levels of the HSF and HSP/heat-response proteins encoding genes are marked in different colors. *Red* represents the significant up-regulated gene expression, log_2_ (fold changes) > 1. *Blue* represents the significant down-regulated gene expression, log_2_ (fold changes) < −1.

**Table 3 plants-11-00677-t003:** Key differentially expressed metabolites in ZD309 and its parental lines in heat treatments.

Treatment	ID	Description	VIP	Fold Change	*t*-Test *p*-Value	Up/Down
D3HS	M248T257	Meperidine	3.8	20.07	1.69 × 10^−13^	up
	M178T176	5-Acetyl-2,3-dihydro-6,7-dimethyl-1H-pyrrolizine	2.32	5.62	2.81 × 10^−10^	up
	M340T240	Acylcarnitine 12:2	2.55	4.97	5.18 × 10^−09^	up
	M261T257	Khellin	1.73	3.21	8.19 × 10^−11^	up
	M364T214	Isopentenyladenine-9-N-glucoside	1.08	2.69	3.53 × 10^−12^	up
	M376T196	Gatifloxacin	1.46	2.69	5.80 × 10^−9^	up
	M220T114	trans-Zeatin	1.13	2.32	0.0003	up
	M273T337	.beta.-Estradiol	1.3	2.25	0.0003	up
	M175T250	1-Methyl-4-(1-methyl-2-propenyl)-benzene	1.12	2.21	0.0007	up
	M355T293	Isoxanthohumol	1.06	2.11	0.0005	up
D6HS	M211T234	Jasmonic acid	2.74	21.7	1.54 × 10^−12^	up
	M193T234	2-Methyl-1-phenyl-2-propanyl acetate	2.81	19.64	1.76 × 10^−10^	up
	M175T234	Glycyl-Valine	2.47	8.62	5.19 × 10^−6^	up
	M112T61	Cytosine	2.47	10.6	2.06 × 10^−6^	up
	M364T214	Isopentenyladenine-9-N-glucoside	2.31	10.3	6.59 × 10^−16^	up
	M150T189	Methionine	2.28	5.44	3.24 × 10^−6^	up
	M213T374	Dihydrojasmonic acid	1.87	4.83	5.14 × 10^−11^	up
	M355T309	Isoxanthohumol	1.68	4.22	1.81 × 10^−10^	up
	M147T195	Coumarin	1.63	3.09	8.10 × 10^−5^	up
	M261T56_2	Mannose 6-phosphate	1.29	2.08	2.13 × 10^−6^	up

## Data Availability

All data generated or analyzed during this study are included in this published article (including its Appendix A).

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
