# Peer review of "Uncovering the Gene Regulatory Network of Maize Hybrid ZD309 under Heat Stress by Transcriptomic and Metabolomic Analysis"

_plants, 2022, doi:10.3390/plants11050677_

Round 1

Reviewer 1 Report

The subject of the article is very interesting, but there are some shortcomings in the work.

  1. In the article, please distinguish between heat stress and drought. These words have a different scientific meaning.
  2. In how many repetitions the experience was founded.
  3. Please describe the tested variety in more detail (company, FAO, type of grain, etc.?)
  4. How were the pots watered?
  5. What was the soil in the pots, what was the chemical composition, NPK content?
  6. Please describe the thermal conditions inside and outside the greenhouse in more detail? 7. How was the biometric features determined. Please describe, for example, the grain yield in more detail
  7. Genetic (cytological) matters are correct
  8. In the keywords, add the grain yield, biometric features
  9. Revise the literature list
  10. In the introduction, add a text on the impact of agrotechnical factors on drought stress, e.g. selection of varieties, mineral fertilization (K), etc.

Author Response

Dear the Editor and Reviewer,

Thank you very much for your critical comments for our manuscript. We have revised the manuscript carefully according to your suggestions. The major revisions are as following,

Response to Reviewer 1

  1. In the article, please distinguish between heat stress and drought. These words have a different scientific meaning.

Yes, heat stress and drought are different. Our present work only investigated the heat stress response and omics alterations. The field and potted maize experiments were well watered to ensure no drought stress.

  1. In how many repetitions the experience was founded.

Three replications were set in potted maize experiments. We have added descriptions in the materials and methods section.

  1. Please describe the tested variety in more detail (company, FAO, type of grain, etc.?)

The summer hybrid of maize, ZD309, is bred by Henan Academy of Agricultural Sciences, China. This dent grain maize hybrid is suitable for mechanized harvesting. We have added the related information in the manuscript.

  1. How were the pots watered?

In this work, the potted maize plants were well watered every two days to avoid water stress. We have added the description in the manuscript.

  1. What was the soil in the pots, what was the chemical composition, NPK content?

The experimental soil belongs to yellow-cinnamon soil, which contains total N for 0.74 g kg-1, available P for 42.6 mg kg-1, 163.0 mg kg-1, and organic matter for 33.0 g kg-1 (PH 8.83). We have added the information in the manuscript.

  1. Please describe the thermal conditions inside and outside the greenhouse in more detail?

We have added the outdoor temperature information and revised the description.

  1. How was the biometric features determined. Please describe, for example, the grain yield in more detail

All the collected data from phenotypic analysis, physiological parameter measurements, and qRT-PCR analysis data were subjected to one-way variance analysis (ANOVA) and Student’s t-test using software SPSS 22.0 (IBM, NY, USA). p < 0.05 indicates the statistical differences to reach the significant different level, p < 0.01 for very significant different levels. We have added a description in the materials and methods section.

  1. Genetic (cytological) matters are correct

The genetic matters of the tested maize inbred lines and hybrid are all correct. The inbred lines, H39_1 and M189, exhibit well genetic and phenotypic stability. The maize hybrid, ZD309, was hand mated by crossing H39_1 with M189.

  1. In the keywords, add the grain yield, biometric features

We have added hybrid, gene regulatory network, grain yield, physical alterations as the keywords.

  1. Revise the literature list

We have checked the literature list carefully and revised the typing errors and added the missing information.

  1. In the introduction, add a text on the impact of agrotechnical factors on drought stress, e.g. selection of varieties, mineral fertilization (K), etc.

The present study mainly focused on maize heat stress. Like drought stress, the effects of heat stress are largely determined by maize variety selection and field conditions. According to the reviewer’s suggestion, we have added a description in the introduction section.

Reviewer 2 Report

The study was focused on unveiling the gene regulatory network of maize hybrid ZD309 under heat tolerance by transcriptomic and metabolomic analysis.

The paper is quite interesting and valuable. However, I recommend some improvements:

  • Figures 1 and 5 – Statistical test results should be added.
  • Description of real-time qRT-PCR analyses should be profoundly extended.
  • In addition, my several experiments evidenced that the expression of GAPDH gene is often unstable under exposure of maize to stressing factors, therefore, I strongly recommend to use another house-keeping gene. For example, actin or tubulin genes.
  • There is the lack of a separate paragraph, describing the conclusions and statistical analyses used.
  • Moderate English changes by the native speaker are required.

Author Response

Dear the Editor and Reviewer,

Thank you very much for your critical comments for our manuscript. We have revised the manuscript carefully according to your suggestions. The major revisions are as following,

Response to Reviewer 2

  1. Figures 1 and 5 – Statistical test results should be added.

We have added the statistical test results in the figures.

  1. Description of real-time qRT-PCR analyses should be profoundly extended.

We have revised the description of qRT-PCR analyses carefully.

  1. In addition, my several experiments evidenced that the expression of GAPDH gene is often unstable under exposure of maize to stressing factors, therefore, I strongly recommend to use another house-keeping gene. For example, actin or tubulin genes.

GAPDH gene is a common internal control of qRT-PCR analysis in plants and animals. However, its expression is usually affected by abiotic stress factors. It is not a good choice used GAPDH gene as the internal control in the present work. We have checked our results in qRT-PCR analysis and the RNA-seq data, all the tested genes exhibited the same trends. It is possible that the effects of heat stress can’t reverse the GAPDH expression trend. Our results are mostly acceptable.

  1. There is the lack of a separate paragraph, describing the conclusions and statistical analyses used.

We have added a separate paragraph, Statistical Analysis, in the materials and methods section.

  1. Moderate English changes by the native speaker are required.

We have revised the English writing carefully, and also asked Dr. Aneirin A. Lott from University of Florida to revise the English writing again.

Reviewer 3 Report

This manuscript presents a detailed transcriptomic and metabolomic analysis of a heat tolerant maize hybrid under heat stress, in order to reveal the respective regulatory network. This study and its results are extremely interesting and useful for maize breeders as well as for other researchers studying the responses of plants to high temperatures. The importance of  this work becomes even greater if we think about the future of agriculture in the new "global warming" era. 

Nevertheless, there are some shortcomings that need to be addressed by the authors. 

Materials and methods:

Page 18, paragraph 4.1, line 469, "seedling leaves": Please define which leaves were sampled and why those leaves were chosen. 

Page 19, paragraph 4.2, lines 475-479: Please indicate which testing kit was used for each determination. Moreover, please provide the website of the company, or any other information about the principle of each kit method. 

Page 19, paragraph 4.3, lines 482-483: Please describe in details the RNA extraction protocol or indicate the respective reference. 

Page 19, paragraph 4.5, line 512: Please provide the Table S1 with the primers' sequences of the DEGs.

Page 19, paragraph 4.6, line 522: What do you mean by saying "20 μL of sample"? Which liquid sample are you referring to?

Results:

Figure 1 (page 4) and Figure 5 (page 9): Please provide statistical analysis and indicate statistical significance p-values.  

Figure 1 (page 4), legend (line 89): Have you determined proline of hydroxyproline content? Please define. 

Figure 1 (page 4): Please describe in details all the respective measurement units. For example, for protein content, what does "mg.ml-1" mean? Or for POD activity, what does "mg-1.pro-1" mean? 

Author Response

Dear the Editor and Reviewer,

Thank you very much for your critical comments for our manuscript. We have revised the manuscript carefully according to your suggestions. The major revisions are as following,

Response to Reviewer 3

  1. Page 18, paragraph 4.1, line 469, "seedling leaves": Please define which leaves were sampled and why those leaves were chosen. 

The upper most whole leaf was selected and sampled. We have revised the improper description.

  1. Page 19, paragraph 4.2, lines 475-479: Please indicate which testing kit was used for each determination. Moreover, please provide the website of the company, or any other information about the principle of each kit method. 

We have provided the related information in the materials and methods sections.

  1. Page 19, paragraph 4.3, lines 482-483: Please describe in details the RNA extraction protocol or indicate the respective reference. 

Total RNA was extracted from the maize leaves of the three materials using Trizol reagent (Invitrogen, CA, USA) according to the manufacturer’s instructions. We have revised this point.

  1. Page 19, paragraph 4.5, line 512: Please provide the Table S1 with the primers' sequences of the DEGs.

We have provided the Table S1 as a supplementary file.

  1. Page 19, paragraph 4.6, line 522: What do you mean by saying "20 μL of sample"? Which liquid sample are you referring to?

We have revised this mistake and cited the corresponding literature.

  1. Figure 1 (page 4) and Figure 5 (page 9): Please provide statistical analysis and indicate statistical significance p-values.  

We have performed statistical analysis and added the statistical significance in Figure 1 and Figure 5.

  1. Figure 1 (page 4), legend (line 89): Have you determined proline of hydroxyproline content? Please define. 

In the present study, proline content was determined. We have revised this point in the manuscript.

  1. Figure 1 (page 4): Please describe in details all the respective measurement units. For example, for protein content, what does "mg.ml-1" mean? Or for POD activity, what does "mg-1.pro-1" mean? 

We have checked all the respective measurement units for the determined physical characters, and revised same mistake. Moreover, we described all the respective measurement units in the figure legend.

Round 2

Reviewer 1 Report

I accept the answers to my questions in the review.

I recommend that you post this article

Author Response

Thank you for your critical comments for our manuscript once again.

Reviewer 3 Report

I would like to thank the authors for their responses to my comments. There still are some shortcomings that need to be addressed by the authors. 

Paragraph 4.2: The provided website of the company is not working. Please check if there is any mistake in page's link. 

Figure 1, units: When you say "g of sample", you mean "g of sample's fresh weight"? Please define.

Figure 1H, units for BCA protein content: I think you mean mg.g-1 and not mg.ml-1 in the graph.

Figure 1M, units for CAT activity (in the graph as well as in figure's legend): μmol is not a measurement unit for enzyme activity. Do you maybe mean μmol O2 .min-1.mg-1? Please check again the principle of your kit method. 

Table S1: Please check again the primers pairs you designed for Zm00001d015300 and Zm00001d052194. Using NCBI primer blast, no target templates were found for Zm00001d015300. Primer pair designed for Zm00001d052194 does not give one specific target template, but rather a number of products with lengths of 90, 93, 219 and 224 bp. 

Author Response

Dear the Reviewer,

Thank you very much for your critical comments for our manuscript once again. We have revised the manuscript carefully according to your suggestions. The major revisions are as following,

Paragraph 4.2: The provided website of the company is not working. Please check if there is any mistake in page's link. 

We have checked the website, it is workable in China. Maybe, there was gateway limited in your country or area.

Figure 1, units: When you say "g of sample", you mean "g of sample's fresh weight"? Please define.

It should be g of sample's fresh weight. We have revised this mistake.

Figure 1H, units for BCA protein content: I think you mean mg.g-1 and not mg.ml-1 in the graph.

Yes, we have revised this mistake.

Figure 1M, units for CAT activity (in the graph as well as in figure's legend): μmol is not a measurement unit for enzyme activity. Do you maybe mean μmol O2 .min-1.mg-1? Please check again the principle of your kit method. 

We have checked the kit method and revised this mistake in the manuscript. The units for CAT activity is U min-1 mg-1, 1 activity unit (U) represents the A240 absorbance decreased 0.1 OD in 1 minute and 1 mg of sample's fresh weight.

Table S1: Please check again the primers pairs you designed for Zm00001d015300 and Zm00001d052194. Using NCBI primer blast, no target templates were found for Zm00001d015300. Primer pair designed for Zm00001d052194 does not give one specific target template, but rather a number of products with lengths of 90, 93, 219 and 224 bp. 

We have checked all the primers. Some gene sequences are different between Gramene and NCBI databases. We designed these primers based on the sequences from the website, https://ensembl.gramene.org/Zea_mays/Info/Index. And all the primers are workable.

Zm00001d015300

>Zm00001d015300_T002 Zm00001d015300:Zm00001d015300_T002 cdna:protein_coding

CACATATAAGCCACCGCCGCCTCGCATCTCCTCCTCCGCCCCACGCAGCTGAGGCGTTGATCCTCACCGGCGTCGCCATCGTAATCGCCTTGGACAGCGGCACCAGAACCTCTGCCTCCGCAGGGTGCTTGCTGGTATCATTACAATCTGGTAGTTTTAGTGGTTGGGCTTGACTTCCTACTATTCAGAAACACAGTACCGAGAAGTCTTTAGGTGGGGTCAAAGATGGTGTCGCTGAAGTTGCAAAAGCGCCTTGCCGCGAGCTACCTCAAGTGTGGGAAAGGCAAGGTGTGGCTTGACCCAAATGAAGTTAGTGAGATCTCCATGGCAAACTCCCGCCAGAATATCCGGAAGTTGGTTAAGGATGGGTTTATCATCAAGAAGCCTCATAAGATCCACTCTAGATCCCGTGCAAGAAGGGCACATGACGCCAAGCAAAAGGGACGCCACTCAGGATATGGTATGCTGCTGCAACTCATTTATGTCACAGTTCTGAAGGGTAAGCGTAGGGGTACCAGGGAGGCTAGGCTACCCACCAAGGTTCTGTGGATGAGGAGGATGCGTGTGCTGAGGCGCCTTCTCCGCAAGTATCGTGAGGCCAAGAAGATTGACAAGCACATGTATCATGACATGTACATGAAGGTCAAAGGTAACGCCTTCAAGAACAAGAGGGTGCTTATGGAGAGTATCCACAAGTCCAAGGCTGAGAAGGCCAGAGAGAAGACACTTTCTGACCAGTTTGAGGCCAAGCGCGCCAAGAGCAAGGCGAGCCGTGAGAGGAAGATCGCTAGGAGGGAGGAGAGGTTGGCCCAGGGTCCACGAGAAATAACACCACCTGTCTCATCCACAGCTCCAGCTACTGGGGCACCGAAGAAAGCAAAGAAGTGAAGTATTGAAATAAAGCTGTAGCTTTCTAGTCCAGTACTGCACTTTCTTTTATAAGCTCTAGTCAAATTTTGGAGCGCCTGAACTGTTTTGCTGTTATATAACTTAGTGTATTGATAAATTCGACTATCACATCTGGATTATTCTGCTGGATAATGTCCGTGTCTTTGATGTTTGTGTTAAATTCCTATTCTGGATG

Zm00001d015300-F  5'  TTCTGAAGGGTAAGCGTAGGG  3'   59.0

Zm00001d015300-R  5'  TCAGCCTTGGACTTGTGGATA  3'   58.0  266-482=217bp

Zm00001d052194

>Zm00001d052194_T001 Zm00001d052194:Zm00001d052194_T001 cdna:protein_coding

ACCGCAAGGATCCACCACAACAGCGAAGGAGAAAGCAGACCAACCTAGCCACCCAGGGAGAAAGAGGCCAAAAGGGAGGGGAGAGTGTCGTCATGGCTTCCATTGTCGCTTCCAGGAGGGCCGTTCCTCTAGTTCGCGCTCTGGAGAAGCTCATCGCAGCGTCCTCCGCTCCCGGGACTGGCTCCGCCCTCAGGCCGGTGGCAGTCGCCGGCGGCCTCCGCGGCTACAACACCGGCGCTCCGCTCCGACGCTACGAGGGGGCCGAGTCGGAAGACGATAGCGTCCGCGAGTACGATGGGCGGCACGGCGGCCGGGACTACGCTGTGCCCAGCCTGTTCTCAGATATTTTCCGTGATCCGCTTAGTGCGCCGCACAGCATTGGCCGCCTGCTGAACCTTGTGGACGACTTGGCGGTGGCGGCGCCAGGTCGTGCGGTGCGCCGTGGCTGGAACGCGAAGGAGGACGAGGAGGCGCTGCACCTGAGGGTGGACATGCCAGGCCTGGGGAAGGAGCACGTCAAGGTGTGGGCGGAGCAGAACAGCCTGGTGATCAAGGGCGAGGGCGAGAAGGAGGATAGCGAGGACGAGGCCGCCCCGCCTCCGAGATACAGCGGTCGCATCGAGCTCGCGCCAGAGGTTTACAGGATGGACAAGATCAAGGCGGAGATGAAGAACGGCGTGCTCAAGGTGGTCGTGCCGAAGGTGAAGGAGCAGCAGCGCAAGGACGTGTTCCAAGTCAACGTCGAGTAGATGTTTCCAAATAGAAGCAAGTGCCGGTACGGGATGGAGGATTGGAGGGGCACTGCCAAACTAGGATTCCTCTCTCTCAATCTGATCTGGATTCTGGAATCAGATTTCTCTTCTTTCATTTTTCTCGTCTATCTTCTATCAGTATGAAATAAGCAACGTCGCTTCAGTTTTCGTGTCAAGGCCGGTGGAGTCGCCTATGTTTATTTTATTTTCTTTGTATTTCCTACCTGGACACACGTTCTCTATGCCGTGTTTGGTTTCCGCAGATTTTTAAAATATGCATGTTCAAACCCAA

Zm00001d052194-F 5' CTGAACCTTGTGGACGACTTG  3' 57.9

Zm00001d052194-R 5' CCCTTGATCACCAGGCTGT 3' 57.7 390-556=167pb

Zm00001d052194-Fn 5' TTCCATTGTCGCTTCCAGG 3' 59.0

Zm00001d052194-Rn 5' GTGCCGCCCATCGTACTC 3' 59.2 98-305=208bp

Zm00001d052194-Fx 5' CTGTGCCCAGCCTGTTCTC 3' 58.5

Zm00001d052194-Rx 5' CCAAGTCGTCCACAAGGTTC 3' 57.5 322-411=90bp (selected)

Round 3

Reviewer 3 Report

I would like to thank the authors for their responses. I believe that the manuscript can be accepted for publication. 

Author Response

Dear the Reviewer,

Thank you very much for your critical comments for our manuscript.